# On-the-Fly Adapting Code Summarization on Trainable Cost-Effective Language Models

**Yufan Cai**
Shanghai Jiao Tong University
National University of Singapore
`cai_yufan@u.nus.edu`

**Yun Lin***
Shanghai Jiao Tong University
`lin_yun@sjtu.edu.cn`

**Chenyan Liu**
National University of Singapore
`chenyan@u.nus.edu`

**Jinglian Wu**
National University of Singapore
`jinglian_wu@u.nus.edu`

**Yifan Zhang**
National University of Singapore
`yfzhang@nus.edu.sg`

**Yiming Liu**
National University of Singapore
`e0945794@u.nus.edu`

**Yeyun Gong**
Microsoft
`yegong@microsoft.com`

**Jin Song Dong**
National University of Singapore
`dcsdjs@nus.edu.sg`

## Abstract

Deep learning models are emerging to summarize source code to comment for code documentation and program comprehension. We can achieve good performance by training the model on large training corpus. However, in practice, the code samples from different projects can have contradictory training signal for learning a deep comment generator, making the model struggled to fit all the training samples.

In this work, we introduce a novel approach, AdaCom, to improve the performance of comment generators by on-the-fly model adaptation. This research is motivated by the observation that deep comment generators often need to strike a balance as they need to fit all the training samples. Specifically, for one certain target code $c$, some training samples $S_p$ could have made more contributions while other samples $S_o$ could have counter effects. However, the traditional fine-tuned models need to fit both $S_p$ and $S_o$ from a global perspective, leading to compromised performance for one certain target code $c$. In this context, we design AdaCom to (1) detect whether the model might have a compromised performance on a target code $c$ and (2) retrieve a few helpful training samples $S_p$ that have contradictory samples in the training dataset and, (3) adapt the model on the fly by re-training the $S_p$ to strengthen the helpful samples and unlearn the harmful samples. Our extensive experiments on 7 comment generators and 4 public datasets show that (1) AdaCom can significantly boost the performance of comment generation (BLEU4 score by on average 14.9%, METEOR by 12.2%, and ROUGE-L by 7.4%), (2) the adaptation on one code sample is cost-effective and acceptable as an on-the-fly solution, and (3) AdaCom can adapt well on out-of-distribution code samples.

## 1 Introduction

Deep learning models are increasingly emerging to generate comments from the source code, facilitating programmers' tasks such as code documentation [27], program comprehension [18], and reverse-engineering [17]. Regarding the code-to-comment generation problem as a language translation or

---

*Corresponding author

37th Conference on Neural Information Processing Systems (NeurIPS 2023).

summarization problem, many deep language models, such as CodeBERT [11], GraphCodeBERT [15], CodeT5 [37], and CodeGPT [14], are designed as state-of-the-art code comment generators. The forthcoming ChatGPT [13] also demonstrates its effectiveness in emulating human-like interpretation of the target code.

On one hand, to improve the model performance, deep language models have evolved to scale up their size. (Ultra-)large language models such as GPT3 [28], CodeX [7], and InstructGPT [29] have shown their effectiveness. However, they usually consist of millions of neurons, which results in substantial training and maintenance costs for individuals.

On the other hand, the relatively small models such as CodeBERT and CodeT5 are more economical and maintainable, but they can struggle to compromise their performance on different subsets of training samples. Our empirical study with the CodeBERT model on the CodeSearchNet[19] dataset shows that on average 49.51% samples are more or less "conflicting" with each other. In other words, if the model only fits some consistent training samples for one certain test case, it will show much better performance. Overall, we observe that the conflicting effects are prevalent, regardless of the types of models and the types of training corpus.

For the sake of a cost-effective comment generator, we propose AdaCom, an on-the-fly model adaptation solution to boost the performance of deep comment generators. We design AdaCom to fulfill three research goals:

- **Compromise Detection**: Given a source code, AdaCom aims to detect whether a comment generator may have a compromised performance on the source code.

- **Samples Searching**: Given a source code and a dataset, AdaCom aims to search for a few samples in the training dataset that can contribute to mitigating the model bias.

- **On-the-Fly Model Tuning**: Given a source code and few labeled samples, AdaCom aims to retrain the model on the fly to boost its performance over the source code.

AdaCom takes input from a target code $c$, a comment generator $g$, and a dataset $D$. AdaCom adapts $g$ on the fly into $g'$ so that $g'$ can have a better performance on generating the comment of $c$. Our rationale lies in that there could be a small subset of samples $D_s \subset D$ which is more helpful for the comment generator to learn to summarize $c$. Technically, we first build an influential graph $G_{inf}$ on the dataset $D$ where each node is a sample and the edge between two nodes indicates the helpful or harmful relation between the two training samples. Secondly, given a target code $c$, we use a model-representation-based measurement to find the potentially semantic helpful samples of $c$ in the training dataset. Then, we enrich the set of helpful samples $D$ based on the $G_{inf}$. Those samples are chosen to fine-tune the model in a lightweight way so that the model can (1) better *learn* helpful samples and (2) *unlearn* harmful samples. Finally, we retrain the model $g$ to $g'$ with those samples on the fly for the target code $c$.

The solution of AdaCom is orthogonal to many deep-learning-based comment generators including improving model architectures [25], model training [15], and code representation [22].

We evaluate AdaCom on boosting 7 comment generators over 4 datasets. The experimental results show that (1) AdaCom can significantly boost the performance of comment generation (the BLEU4 score by on average 14.9%, METEOR by 12.2% and ROUGE-L by 7.4%), (2) the whole adaptation on an individual code sample takes a very small run-time overhead, and (3) AdaCom can generalize well towards out-of-distribution code samples.

## 2 Motivating Example

Table 1 shows an example of how AdaCom applies on-the-fly adaptation to adapt the comment generator to improve its performance. In the table, the fourth row shows the origin-generated comment after model fine-tuning, the fifth row shows the generated comment after model trained by the helpful training samples, and the last row shows the generated comment after model trained by the harmful training samples.

**Step 1. Influence Construction.** In this example, AdaCom first establishes the influential relationship among all the training samples. For clarity, we only show the influence between $c_{harm}$ and $c_{help}$

Table 1: The motivating example to illustrate the compromise of the comment generator. The difference is shown on the last three rows (i.e., model prediction after trained by both $c_{harm}$ and $c_{help}$, after retrained by $c_{harm}$, after retrained by $c_{help}$.

| Sample | Target Test Sample, $c_t$ | Harmful Training Sample, $c_{harm}$ | Helpful Training Sample, $c_{help}$ |
|---|---|---|---|
| Code | ```public static java.sql.Timestamp internalToTimestamp(long v) { return new java.sql.Timestamp (v - LOCAL_TZ.getOffset(v)); }``` | ```private Date longToDate(long val, int sqlDataType) { switch (sqlDataType) { case Types.DATE: return new java.sql.Date(val); case Types.TIME: return new java.sql.Time(val); case Types.TIMESTAMP: return new java.sql.Timestamp(val); } }``` | ```public static java.sql.Date internalToDate(int v) { final long t = v * MILLIS_PER_DAY; return new java.sql.Date(t - LOCAL_TZ.getOffset(t)); }``` |
| Ground Truth | Converts the internal representation of a SQL TIMESTAMP (long) to the Java type used for UDF | Parse the long-valued timestamp into the appropriate SQL date type | Converts the internal representation of a SQL DATE (int) to the Java type used for UDF parameters (@link java.sql.Date) |
| Origin | Converts the internal { @ link long } ( local to a { @ link TIMESTAMP } ) representation | | |
| After Harmful | Parse the long-valued TIMESTAMP into a { @ link TIMESTAMP } representation | | |
| After Helpful | Converts the internal representation of a SQL TIMESTAMP ( long ) to the java type used for UDF | | |

Table 2: Influence and estimated training contribution for examples in Table 1

| Sample | Estimated Influence | | Estimated Contribution | |
|---|---|---|---|---|
| | $c_{harm}$ | $c_{help}$ | $c_{harm}$ | $c_{help}$ |
| $c_t$ | / | / | 0.23 | 0.67 |
| $c_{harm}$ | 1 | -0.76 | / | / |
| $c_{help}$ | -0.76 | 1 | / | / |

in Table 2. Generally, we can see that $c_{harm}$ and $c_{help}$ have an estimated influence score of -0.76, indicating that the deep-learning model has made a compromise to fit the pair. We will illustrate the calculation of the influence score in Section 3. In this stage, we define and cache all the "contradictory" pairs in the training dataset.

**Step 2. Training Contribution Construction.** Next, we estimate the potential training contribution of each training sample to the target code $c_t$ and select $c_{help}$ as one of the semantic contributing training samples. Intuitively, $c_t$ and $c_{help}$ shares a similar representation within the deep comment generator, which serves as a clue for us to select $c_{help}$. We further enhance the $c_{help}$ to a set of code samples using the influence score. More details can be found in Section 3.2. In this stage, we construct the possible helpful samples from the training dataset.

**Step 3. On-the-fly Retraining.** Finally, helpful code samples like $\{c_{help}\}$ are used to retrain the comment generator $g$ to $g'$. In Table 1, both $c_{harm}$ and $c_{help}$ are well fitted by the model. However, we can observe the changes of generated comments in the last three rows of Table 1 when retraining happens. The origin comment generator generates a more general and inaccurate comment but after re-training with the helpful sample, the model generates a more precise comment for $c_t$.

Overall, AdaCom is designed for finding possible compromises and adapting the model under specific scenarios. Note that, both $g$ and $g'$ make a compromise. The original comment generator $g$ inclines to favor fitting both $c_{harm}$ and $c_{help}$, which optimizes $g$ from a global perspective. In contrast, $g'$ inclines to favor fitting $c_{help}$ over $c_{harm}$, which optimizes $g'$ locally for improving the comments for the target code $c_t$. As shown in our experiment (see Section 4), the adaptation boosts the performance of state-of-the-art comment generators with acceptable run-time overhead.

# 3 Approach

Figure 1 shows an overview of the design AdaCom, which consists of an offline stage and an online stage. In the offline stage, a comment generator $g$ is assumed to be trained from the training dataset $D$. Based on $D$ and $g$, we define the influence estimation and construct the influence graph among the training samples in $D$ by their pairwise mutual influence. We cache both the representation of the training samples and the influence scores for online retrieval.

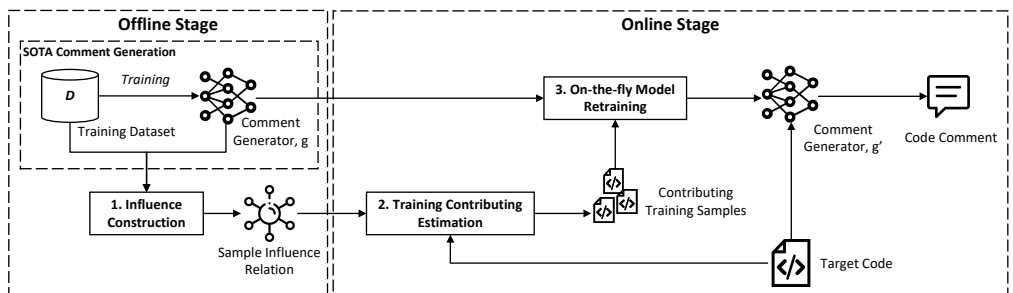

Figure 1: Overview of AdaCom. We assume a state-of-the-art approach has been adopted to train the comment generator $g$ on the dataset $D$. Influence Construction in the offline stage is for characterizing the helpful and harmful samples and training contribution estimation in the online stage is for retrieving the semantic helpful samples.

In the online stage, we retrain the comment generator $g$ to $g'$ with the retrieved contributing training samples and then generate the new code comment by $g'$.

## 3.1 Influence Construction

We introduce a gradient-based estimation for efficiently calculating *model-dependent* mutual influence of any pairs of training samples.

**Influence Estimation**   We denote a comment generator as $g$, its training dataset as $D = \{d_1, d_2, ..., d_n\}$ where $d_i = (c_i, com_i)$, $c_i$ is the code and $com_i$ is its comment. Let $\mathcal{L}(g(c_i), com_i)$ be the loss of $d_i$, and the parameters of the trained comment generator $g^*$ as $\theta$, we calculate the empirical mutual influence $mul\_inf_e$ of $d_1$ and $d_2$ as:

$$mul\_inf_e(d_1, d_2) = \frac{grad(d_1) \cdot grad(d_2)}{|grad(d_1)| \cdot |grad(d_2)|} \tag{1}$$

$$grad(d_i) = \frac{\partial \mathcal{L}(g^*(c_i), com_i)}{\partial \theta} \tag{2}$$

In another word, we now empirically investigate how likely we can reduce the loss of $d_1$ by reducing the loss on $d_2$ (and vice versa) on the comment generator $g^*$. Specifically, $mul\_inf_e(d_1, d_2)$ being close to 1 indicates that $d_1$ and $d_2$ are mutually helpful, that being close to -1 indicates that $d_1$ and $d_2$ are mutually harmful and that being 0 indicates that $d_1$ and $d_2$ are independent.

**Influence-based Graph**   We further build the influence graph by calculating the mutual influence between every pair of the training samples. Generally, the resultant influence graph serves as a cache to improve the retrieval efficiency of an unseen target code.

## 3.2 Training Contribution Construction

In this section, given a target code $c_t$ with unknown comment, a training sample $c_{tra}$ with labelled comment $com_{tra}$, and a comment generator $g$, we estimate how likely $d_{tra} = (c_{tra}, com_{tra})$ can be used to retrain $g$ to improve the generated comment for $c_t$.

**Semantic Similarity**   Our rationale lies in that, from the perspective of the model $g$, if the training samples have a more similar internal representation with that of $c_t$, they are more likely to share the similar distribution of $c_t$, thus having more semantic informative to be contributing to predict $c_t$.

**Representation Assumption**   In this work, we assume that the deep-learning-based comment generators have an internal representation of each code sample. We denote a code sample as $c = \langle t_1, t_2, ..., t_n \rangle$ and the comment generator as $g$. Then, $g$ can generate an internal representation when parsing $t_i$, denoted as $r_i = h(t_i)$, where $h(.)$ is a representation function inside $g$. This

assumption generally holds for most neural network-based approaches. For example, all one-directional or bi-directional recurrent neural networks and transformer-based model architectures conform to the assumption.

**Contribution Estimation**   Given the code sample $c = \langle t_1, t_2, ..., t_n \rangle$, we can derive its sequence of internal representation as $\hat{c} = \langle h(t_1), h(t_2), ..., h(t_n) \rangle = \langle r_1, r_2, ..., r_n \rangle$. Since each code will be represented by a sequence of high-dimensional vectors, we compare the target code $c = \langle r_1, r_2, ..., r_n \rangle$ with all training samples. In this work, we score training contributions by calculating the cosine similarity between the vectors of two code samples' internal representations.

### 3.3   On-the-fly Model Retraining

Given a target code $c$ and two thresholds $th_1$, $th_2$, AdaCom first retrieves a few samples $M = \{m_1, m_2, ...\}$ from all the training samples whose training contribution score is higher than the user-defined $th_1$. If no training samples can be retrieved, we do not retrain the model. Otherwise, AdaCom retrieves some new samples $N = \{n_1, n_2, ...\}$ from the influential graph related to nodes in $M$ (see Section 3.1). The estimated influence score between the samples in $M$ and the samples in $N$ should be higher than the user-defined $th_2$. The threshold $th_1$ is designed to filter out the semantic helpful training samples and the threshold $th_2$ is to find out the samples that could help the model unlearn the mutually harmful samples during the learning of some mutually helpful samples. If no training samples are mutually harmful to the samples in M and N based on the influence graph, AdaCom will remove these samples for further retraining. Finally, $g$ is retrained to $g'$ by all the retrieved samples and then generates the new comment as $g'(c)$.

For the sake of run-time overhead, we freeze the majority of the neural network and retrain only the last few layers. We also adopt a dropout training strategy and early stop mechanism to remedy the over-fitting and run-time overhead problem.

## 4   Evaluation

We evaluate AdaCom with the following research questions:

- **RQ1**: Whether AdaCom can boost the performance of diverse comment generators?
- **RQ2**: Whether AdaCom can boost the performance of comment generation on diverse datasets?
- **RQ3**: Whether AdaCom can outperform the retrieval-augmented neuron-based approaches?
- **RQ4**: How generalized AdaCom can boost the performance on the out-of-distribution samples?
- **RQ5**: What is the runtime overhead of AdaCom to boost the performance?
- **RQ6**: How each component of AdaCom can contribute to its boosting performance?

**Experiment Setup**   Our experiments are conducted on 2 Ubuntu 20.04 servers equipped with 2 AMD Ryzen[TM] 9 5950X 16-Core CPU, 128 GB memory, and 2 Nvidia RTX[TM] A4000 GPU cards. All the experiment details and replication details can be referred to our website [6].

In this experiment, we adopt four public datasets including CodeSearchNet [19] (CSN), CodeKG [9], FunCom [21] and CosBench [41]. We select the datasets for their diversity and representativeness in terms of cross-language features (CodeSearchNet), available project information (CodeKG), and scalability (FunCom and CosBench). Table 3 shows the details.

We choose seven popular language models including Roberta [26], CodeBERT(small and base version) [11], GraphCodeBERT [15], and CodeT5[37](small, based, and large version). Those models are chosen for their diversity in their pretrained corpus (e.g., Roberta on natural language and CodeBERT on code), architectures, and scalability.

Following existing literature [3, 21, 1, 39, 20, 2, 5], we use smoothing BLEU4 [30][24], METEOR [4], and ROUGE-L [23] to evaluate the performance of code comment generation.

**Experiment Design**   To answer RQ1 (cross-model evaluation), we fine-tune all the language models on the CodeKG dataset. We select the CodeKG dataset for its diversity in the selected projects,

Table 3: Experiment Datasets

| Dataset | Train | Valid | Test |
|---|---|---|---|
| FunCom | 1,954,807 | 104,273 | 90,908 |
| CosBench | 296,425 | 42,348 | 84,694 |
| CodeKG | 161,857 | 20,282 | 40,512 |
| CSN-Python | 251,820 | 13,914 | 14,918 |
| CSN-PHP | 241,241 | 12,982 | 14,014 |
| CSN-Go | 167,288 | 7,325 | 8,122 |
| CSN-Java | 164,923 | 5,183 | 10,955 |
| CSN-JavaScript | 58,025 | 3,885 | 3,291 |
| CSN-Ruby | 24,927 | 1,400 | 1,261 |

and explicit project and code relation information for further analysis [9]. Given the randomness introduced by the dropout strategy in the on-the-fly retraining, we repeat the experiment five times to evaluate the consistency of AdaCom.

To answer RQ2 (cross-dataset evaluation), we evaluate how AdaCom can boost CodeT5-small on all four datasets. We fine-tune the codeT5-small model for each dataset and then apply the AdaCom to all four test datasets.

To answer RQ3 (retrieval-methods comparison), we compare AdaCom with the retrieval-augmented method Retro [16] and the method that uses [CLS] with cosine similarity for helpful examples retrieval. Compared to Retro, AdaCom does not need to retrieve and concatenate similar samples in the training stage but retrieves some samples for each test sample in the test stage.

To answer RQ4 (generalization), we follow the setting in the [35] and create a new dataset split into source and target. The source and target dataset have different *types* of training and testing samples. We evaluate the performance of AdaCom on the model on the target testing dataset by finding helpful samples in the target training dataset. Note that we do not fine-tune the model on the target domain but only cache the estimated influence and representation of the target training dataset in the offline stage. In this case, we can evaluate how AdaCom adapts the model to new domains by only on-the-fly training the helpful samples without the need to fine-tune the whole dataset. We evaluate the generalizability of AdaCom by defining the split-type regarding three different granularity:

- **Cross Languages:** We apply AdaCom to T5 [31] and Roberta model that are pre-trained only on natural language corpus (as source training dataset) to generate code comments on the CodeKG dataset. We evaluate the boosting performance of AdaCom on the CodeKG as the target testing dataset by on-the-fly retraining a few helpful samples in the training dataset of CodeKG.

- **Cross Program Languages:** We apply the AdaCom to enforce the Roberta model trained on CSN-Java to generate code comments on the Python, PHP, Go, Javascript, and Ruby programming languages on the CSN dataset. We train the Roberta model on training data of CSN-Java (source training dataset) and then evaluate the boosting performance of AdaCom on the testing dataset of other programming languages by finding the helpful subset in their training dataset.

- **Cross Projects:** Finally, based on the code project information reported in CodeKG, we split the code corpus by projects, denoted as $D_{pj_1}$ and $D_{pj_2}$. Similarly, we apply AdaCom to enforce the Roberta model trained on $D_{pj_1}$ to generate code comments on $D_{pj_2}$.

To answer RQ5 (runtime overhead evaluation), we instrument the runtime overhead on the experiment for RQ2. We conducted two settings: one utilizing the RTX 3080 GPU on a Windows platform to emulate a programmer's working environment, and another using the A4000 GPU on Ubuntu within the lab setting. We record the average boosting of smoothing BLEU-4 score over all datasets and the consuming time for the whole testing stage. Then we use the boosting smoothing BLEU-4 score per second as the metric to measure the cost performance of different methods.

To answer RQ6 (ablation study), we compare the AdaCom performance with three settings. The first setting just uses the comment of the most helpful sample as the prediction, and the second setting uses the standalone model (i.e., adaptation disabled) to predict the comment. Last, we compare the larger model – CodeT5+[36] with the CodeT5-small and base model equipped with AdaCom.

Table 4: Cross-model evaluation for the boosting performance of AdaCom.

| Model | Scale | Parameter | BLEU4 | | | METEOR | | | ROUGE-L | | |
|---|---|---|---|---|---|---|---|---|---|---|---|
| | | | before | after | bst (%) | before | after | bst (%) | before | after | bst (%) |
| codeT5-small | small | 60M | 34.89±0.26 | 49.05±0.38 | 40.6 | 43.74±0.00 | 56.73±0.08 | 29.7 | 50.98±0.00 | 61.23±0.09 | 20.1 |
| CodeBERT-small | | 84M | 40.83±0.31 | 48.84±0.72 | 19.6 | 48.84±0.32 | 57.07±0.50 | 16.9 | 54.82±0.28 | 60.40±0.47 | 10.2 |
| RoBERTa | base | 173M | 44.73±0.08 | 48.71±0.45 | 8.9 | 52.67±0.17 | 57.23±0.60 | 8.7 | 57.78±0.15 | 60.28±0.59 | 4.3 |
| CodeBERT-base | | 173M | 44.30±0.34 | 48.35±0.43 | 9.1 | 52.35±0.38 | 56.74±0.26 | 8.4 | 57.71±0.39 | 59.81±0.26 | 3.6 |
| GraphCodeBERT | | 173M | 45.51±0.48 | 49.40±0.53 | 8.5 | 53.68±0.58 | 57.82±0.46 | 7.7 | 58.89±0.60 | 61.13±0.48 | 3.8 |
| codeT5-base | | 223M | 45.53±0.00 | 49.79±0.03 | 9.4 | 54.19±0.00 | 57.63±0.15 | 6.3 | 58.48±0.00 | 61.85±0.09 | 5.8 |
| codeT5-large | large | 738M | 45.99±0.65 | 49.87±0.66 | 8.4 | 54.09±0.69 | 58.27±0.47 | 7.7 | 59.29±0.57 | 61.73±0.46 | 4.1 |

Table 5: Cross-dataset evaluation for the boosting performance of AdaCom.

| Dataset | BLEU4 | | | METEOR | | | ROUGE-L | | |
|---|---|---|---|---|---|---|---|---|---|
| | before | after | bst (%) | before | after | bst (%) | before | after | bst (%) |
| CodeKG | 34.89 | 49.05 | 40.6 | 43.74 | 56.73 | 29.7 | 50.98 | 61.23 | 20.1 |
| Cosbench | 29.22 | 31.31 | 7.15 | 35.86 | 37.07 | 3.37 | 37.23 | 37.48 | 0.67 |
| FunCom | 33.32 | 33.76 | 1.32 | 41.71 | 42.04 | 0.79 | 49.23 | 49.53 | 0.61 |
| CSN-java | 19.17 | 20.06 | 4.64 | 32.09 | 32.84 | 2.34 | 38.28 | 38.88 | 1.57 |
| CSN-js | 15.15 | 17.06 | 12.61 | 22.84 | 24.40 | 6.83 | 30.38 | 31.18 | 2.63 |
| CSN-python | 19.71 | 19.92 | 1.07 | 30.57 | 30.68 | 0.36 | 37.23 | 37.48 | 0.67 |
| CSN-go | 18.88 | 19.15 | 1.43 | 33.95 | 34.10 | 0.44 | 41.21 | 41.40 | 0.46 |
| CSN-php | 24.70 | 25.76 | 4.29 | 36.34 | 36.96 | 1.71 | 44.53 | 45.40 | 1.95 |
| CSN-ruby | 14.78 | 15.03 | 1.69 | 25.11 | 25.05 | -0.24 | 31.77 | 31.94 | 0.54 |

**RQ1: Cross-model Evaluation** Table 4 shows that AdaCom can improve the state-of-the-art comment generators with decent significance and consistency. Overall, AdaCom can boost the smoothing BLEU-4 score by on average 14.9%, the METEOR score by 12.2% , and the ROUGE-L score by 7.4%. Compared to the boosted performance in large models, the improvement of AdaCom is more significant on the smaller models, which aligns with the expectation that smaller models can be more "struggled" over conflicting subsets of code samples. In addition, AdaCom performs in a consistent manner in the experiment. The deviation generally ranges between 0 and 0.72 in BLEU4, 0.08 and 0.5 in METEOR, and 0.09 and 0.59 in ROUGE-L.

**RQ2: Cross-dataset Evaluation** Table 5 shows that AdaCom generally works well on different code datasets based on CodeT5-small, which improves on average 8.3% in BLEU4 score, 5.0% in METEOR, and 3.2% in ROUGE-L. We first notice that the performance is not consistent over all dataset. The AdaCom shows much better boosting performance on the CodeKG, Cosbench, CSN-js dataset. Our investigation on these datasets shows that the number of helpful training samples in some dataset such as `CSN-python` are limited, which lead to minor improvement on the model performance. Secondly, as we mainly adjust our hyper-parameters for optimizing BLEU score, the BLEU4 scores increase much higher than other two measurements. We observe that the choice of a stricter threshold of helpful sample selection is a useful and practical mitigation. Given a higher threshold, AdaCom may include less irrelevant training samples to force the on-the-fly retraining. More experiment results on other sizes of models can be referred in our website [6].

**RQ3: Retrieval-methods Comparison** Table 6 shows that AdaCom works generally better than the retrieval-augmented methods. We find that for traditional retrieval-augmented methods, they can not adapt the model to one certain sample simply using the similar code comment. In the training stage, their model still suffer the bias of the contradictory samples and simply concatenate some similar training samples will not avoid the compromise. In the testing stage, AdaCom can adapt the model to re-learn the helpful samples and unlearn the harmful samples to gain a better performance for one certain test sample.

**RQ4: Generalization Evaluation** Table 7 shows the generalizability of AdaCom, which indicates that its boosting performance for cross-language, cross-PL, and cross-project is significant. For adapting from natural language to programming language, both T5 and Roberta model have much better performance even approaching the performance of traditional fine tuning model. In AdaCom, the large language model can skip the time-consuming fine tuning stage but only search few helpful samples in the target dataset and use AdaCom to fir for one certain test sample. For adapting from one

Table 6: The performance of retrieval-augmented method Retro and using CLS embedding with cosine similarity based on CodeT5-base model.

| Dataset | Retro 223M | CLS-Cosine 223M | AdaCom 223M |
|---|---|---|---|
| CSN-java | 20.05 | 20.14 | 20.85 |
| CSN-js | 16.15 | 17.35 | 18.81 |
| CSN-python | 19.67 | 19.58 | 20.46 |
| CSN-go | 19.46 | 19.12 | 19.61 |
| CSN-php | 24.91 | 26.45 | 26.90 |
| CSN-ruby | 14.91 | 14.45 | 15.39 |
| overall | 19.19 | 19.52 | 20.34 |

Table 7: Generalizability of AdaCom on cross-language, cross programming-language, and cross-project evaluation.

| Generalization Type | Option | BLEU4 | | | METEOR | | | ROUGE-L | | |
|---|---|---|---|---|---|---|---|---|---|---|
| | | before | after | bst (%) | before | after | bst (%) | before | after | bst (%) |
| Cross language | T5-base | 10.44 | 41.97 | 302.01 | 24.91 | 57.27 | 129.91 | 18.99 | 50.72 | 167.09 |
| | Roberta-base | 6.42 | 37.51 | 484.27 | 10.24 | 42.85 | 318.46 | 11.45 | 38.77 | 238.60 |
| Cross PL | CSN-JS | 7.00 | 15.50 | 121.43 | 14.37 | 23.48 | 63.40 | 7.90 | 18.16 | 129.87 |
| | CSN-Python | 7.42 | 12.91 | 73.99 | 16.34 | 22.61 | 38.37 | 8.54 | 16.71 | 95.67 |
| | CSN-Go | 4.20 | 11.32 | 169.52 | 9.95 | 21.99 | 121.01 | 4.91 | 17.14 | 249.08 |
| | CSN-PHP | 7.56 | 16.71 | 121.03 | 16.39 | 28.61 | 74.56 | 8.66 | 22.38 | 158.43 |
| | CSN-Ruby | 7.71 | 9.95 | 29.05 | 16.18 | 20.18 | 24.72 | 8.63 | 13.69 | 58.63 |
| Cross project | CodeKG | 11.52 | 35.19 | 205.47 | 20.82 | 45.57 | 118.88 | 28.03 | 48.64 | 73.53 |

programming language to another, AdaCom also shows effectiveness on cross programming language generalizability. For adapting across projects, AdaCom still shows quite competitive performance. The model improves a lot on the out-of-distribution data - the code from unseen projects. Generally, AdaCom has shown its effectiveness to address the distribution-shift problem on diverse scenarios.

**RQ5: Runtime Overhead Evaluation** Table 8 shows the detailed runtime overhead over boosting of smoothing BLEU-4 score using CodeT5-base model as the baseline model. We can see that on Windows, both AdaCom-small and base outperform the CodeT5+ and on Ubuntu, the AdaCom-base shows better performance than CodeT5+. Based on this experiment, we deem that AdaCom can be well applied for the practical performance boost for the cost-effective smaller models.

**RQ6: Ablation study** Table 9 shows that simply utilizing helpful comments or models will decrease the AdaCom performance. In contrast, AdaCom (the last two column) can combine them and adapt the model by the helpful samples to increase the model performance. AdaCom can also adapt the base model to outperform the CodeT5+ large model.

## 5 Related Work

**Large Language Model** Generally, the neural network model architecture follows an encoder-decoder structure. Regarding the code text as sequential data, CNN [2], LSTM [20, 38], or GRU [21] are used to serve as backbone model. With the advance of model, attention mechanism [34] and transformer [1, 8] are adopted to further improve the performance. CodeBERT [11] is a bimodal pre-trained model for multiple programming languages (PLs) and natural language (NLs), it supports

Table 8: Run-time Overhead over BLEU-4 score boosting compared to CodeT5 model.

| | Model Name | CodeT5 | CodeT5+ | AdaCom-small | Adacom-base |
|---|---|---|---|---|---|
| | Parameter Size | 223M | 770M | 60M | 223M |
| Windows | Time (second) | 1.21 | 4.06 | 2.34 | 4.81 |
| | Time per sample (second) | 0 | 2.85 | 1.13 | 3.60 |
| | Average boost BLEU score | 0 | 0.63 | 0.45 | 1.28 |
| | Boost BLEU score per second | - | 22.11% | 39.82% | 35.56% |
| Ubuntu | Time (second) | 0.38 | 1.91 | 1.46 | 3.16 |
| | Time per sample (second) | 0 | 1.53 | 1.08 | 2.78 |
| | Average boost BLEU score | 0 | 0.63 | 0.45 | 1.28 |
| | Boost BLEU score per second | - | 41.18% | 41.67% | 46.04% |

Table 9: Ablation Study on AdaCom

| | Retrieved Comment | Model Only | Larger Model | AdaCom-small | AdaCom-base |
| --- | --- | --- | --- | --- | --- |
| | 223M | 223M | 770M | 60M | 223M |
| CSN-java | 13.18 | 19.17 | 20.83 | 20.06 | 20.85 |
| CSN-js | 13.96 | 15.15 | 17.93 | 17.06 | 18.81 |
| CSN-python | 11.70 | 19.71 | 20.47 | 19.92 | 20.46 |
| CSN-go | 11.17 | 18.88 | 19.64 | 19.15 | 19.61 |
| CSN-php | 17.47 | 24.70 | 26.39 | 25.76 | 26.90 |
| CSN-ruby | 8.91 | 14.78 | 15.63 | 15.03 | 15.39 |
| overall | 12.73 | 18.73 | 20.15 | 19.50 | 20.34 |

downstream NL-PL tasks like code search and code comment generation. Compared with CodeBERT, the input of GraphCodeBERT includes both code tokens and a code data flow graph among the variables. Compared to CodeBERT and GraphCodeBERT, CodeT5 [37] aims at a generic representation of both PLs and NLs. Following the original T5 model [31], the architecture of CodeT5 is faithful to the original Transformer.

**Retrieval Based Comment Generator**   As code duplication is common in large-scale repositories, retrieval-based techniques like vector space model [17] and code clone detection [40] are used to find the most similar code comment pairs from the database in the early stage. By manually designing a set of rules, they can filter out the most similar pair and copy its comment as the output. Recent works combine the retrieval system and neural network model to generate comments. Researchers use neural network as semantic feature extractor to retrieve the comment of the most similar training sample. CCGIR [42] adopts CodeBERT and BERT-whitening operation [32] for retrieval. Re$^2$Com [39] uses BM25 to find the most semantically similar sample and utilizes four encoders to process target code, similar code, similar code's comment, and code abstract syntax tree. Through the attention mechanism, the four encoders' outputs are fused and fed into the decoder to get the final comment. Similarly, in [44], a fusion layer is designed to combine the CodeBERT output of both target code and retrieved code.

Different from the previous works that retrieval during the training stage, we search for the helpful samples in the testing stage to reduce the bias of the model.

**Test-time Adaptation**   Adaptation means the generalization from one distribution to another distribution. Transfer learning [10, 43] is a technique to transfer the parameters of a pre-trained model to a new model on the target dataset. Through transfer learning, we can share the learned parameters (also the knowledge learned by the model) with the new model to speed up the learning procedure. The model is first pre-trained on the source dataset and then trained on the training data of the target dataset. Then it can be used for predicting the test data of the target. Domain adaptation techniques [12] aim to improve the performance of the model in another domain (target domain) to approach the effect of the original domain (source domain). Further, the test-time training technique [33] updates the model during testing with unlabeled data in a self-supervised way. Compared to test-time training, Wang et al. propose [35] an adaptation technique independent of the training data and training loss. The model is trained by test entropy minimization during the test stage and adapts itself to feedback from its predictions.

To the best of our knowledge, we are the first to introduce test-time adaptation for comment generation. Different from the aforementioned works, AdaCom selects the helpful samples from the existing training samples and then adapts the model for each target code sample.

## 6   Limitation

**Scalability** AdaCom requires more time to adapt the model, especially for a billion-size model. To mitigate this, freezing techniques are necessary, specifically freezing the parameters in the encoder and only training part of the parameters of the decoder.

**Construct Validity** AdaCom assumes that it can find helpful examples in the training dataset based on the two metrics to enhance the performance of the model. However, there exist test cases with no helpful examples or possible helpful examples that are ignored. It also remains a question whether our proposed methods would introduce some harmful samples.

Table 10: An example showing the potentially over-fitting problem of AdaCom

| Sample | Target Test Sample, $c_t$ | Harmful Training Sample, $c_{charm}$ | Helpful Training Sample, $c_{help}$ |
|---|---|---|---|
| Code | ```
ReceiptViewModel purchase
    (...) {
    Db.User user = Db.
        getInstance().
        findUserByUserName
        (userName);
    if (user == null) {
        ...
    }
    Db.Account account =
        findAccount(user)
        ;
    return purchase(user,
        account, itemName
        );
}
``` | ```
String receiveRequest(
    Object... parameters)
    throws
    DbUnavailableException
    {
    var id = generateId();
    var req = new
        PaymentRequest(id
        , (float)
        parameters[0]);
    return updateDb(req);
}
``` | ```
ReceiptViewModel purchase
    (...) {
    Db.Product item = Db.get
        ().find(itemName);
    if (item == null) {
        ...
    }
    Receipt receipt = ...;
    if (transaction == null)
        {
        ...
    }
    return receipt;
}
``` |
| Ground Truth | domain purchase with userName and itemName, with validation for userName | public method which will receive request from @link com. iluwatar. commander. Commander | domain purchase with user, account and item-Name, with validation for whether product is out of stock and whether user has insufficient funds in the account |
| Origin | (BLEU 16.78) register purchase with user name | | |
| AdaCom Epoch8 | (BLEU 66.43) domain purchase with user name and item name , with validation for whether user is enabled or not | | |
| AdaCom Epoch11 | (BLEU 84.84) domain purchase with user name and item name , with validation for the account | | |
| AdaCom Epoch15 | (BLEU 47.79) domain purchase with user , account and item name , with validation for whether user is enabled or not | | |

**Internal Validity** We further analyze the cases where the AdaCom shows limited performance. We find that AdaCom sometimes overfits the retrained training samples. In Table 10, despite that AdaCom can successfully select a useful reference $c_{help}$, it cannot stop at a correct point. In this example, AdaCom retrains the deep comment generator for 15 epochs. However, the best stopping condition happens at epoch 11, which means that AdaCom should have stopped earlier to gain higher accuracy and lower run time. We think AdaCom can leave these candidates to the users to choose and use the helpful samples to interpret the model behavior.

## 7 Conclusion

In summary, we propose a novel system - AdaCom which boosts the performance of comment generators on-the-fly. Extensive experiments has shown that AdaCom can effectively improve the performance of the code comment generation on diverse datasets, programming languages, and different deep language models. Further, AdaCom also shows its decent generalizability on cross-language, cross programming-language and cross-project settings. Overall, our solution shows promising results in enhancing model performance and potentially efficiency.

## Acknowledgments and Disclosure of Funding

This research is supported in part by the Minister of Education, Singapore (T2EP20120-0019, MOET32020-0004), NUS-NCS Joint Laboratory for Cyber Security, Singapore, the National Research Foundation, Singapore, and Cyber Security Agency of Singapore under its National Cybersecurity Research and Development Programme (Award No. NRF-NCR_TAU_2021-0002) and A*STAR, CISCO Systems (USA) Pte. Ltd and National University of Singapore under its Cisco-NUS Accelerated Digital Economy Corporate Laboratory (Award I21001E0002), National Research Foundation, Singapore, and the Cyber Security Agency under its National Cybersecurity R&D Programme (NCRP25-P04-TAICeN), and National Natural Science Foundation of China (62172099, 23Z990203011). Any opinions, findings and conclusions or recommendations expressed in this material are those of the author(s) and do not reflect the views of National Research Foundation, Singapore and Cyber Security Agency of Singapore.

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
