# OpenReview forum: "On-the-Fly Adapting Code Summarization on Trainable Cost-Effective Language Models"
_NeurIPS.cc/2023/Conference — NeurIPS 2023 poster_

### Official Review · Reviewer_Ugvn · 2023-07-05

**Soundness:** 3 good
**Presentation:** 3 good
**Contribution:** 2 fair
**Rating:** 5
**Confidence:** 4

**Summary:**

This paper proposes an AdaCom approach to improve the code comment generation performance by on-the-fly model adaptation. For one code sample at test time, AdaCom first detects whether the model might have a compromised summarization performance and if yes,  re-adapt the model on-the-fly by training the most contributing training samples to improve its performance for this sample. By experimenting with 7 deep comment generators and 4 public datasets, results show that AdaCom can effectively improve the performance of the code comment generation on diverse datasets, programming languages (PLs), and code language models (LMs). However, AdaCom achieves the performance improvements at a cost of adding an on-the-fly retraining stage in test time, incurring considerable runtime overhead. Its method seems to be over-complicated compared to the popular retrieval-augmented generation techniques, and there is no comparison with these techniques. This work also lacks a comparison with many state-of-the-art (SoTA) code LLMs such as CodeGen and Incoder.

**Strengths:**

* The proposed AdaCom is technically sound and back with convincing performance gains after adopting AdaCom on multiple datasets covering various PLs and code LMs.
* The paper is well structured and easy to read.

**Weaknesses:**

* While the proposed method boosts the performance of code comment generation, it achieves this at a huge cost of runtime overhead during inference. Though the paper uses some techniques such as model parameter freezing to optimize the runtime cost due to the on-the-fly retraining, it still takes 5.54 seconds for large-size models (below 1B) to generate a comment on average. Such a latency is not that acceptable for real-world applications such as in an integrated development environment, where developers typically expect less than 1 second latency.
* Additionally, the proposed method seems to be over-complicated compared to other widely adopted retrieval-augmented generation techniques [1] that similarly improve performance by referring to some related training examples. There is no comparison with these techniques in terms of both performance and runtime overhead. Though retrieval-augmented generation techniques might require an extra stage of retrieval during training, they seem to be much more efficient during inference which often has a high requirement for low latency. Particularly, in the era of LLMs such as ChatGPT, retrieval-augmented generation becomes a more popular approach to provide retrieved examples for LLMs via in-context learning without any retraining. It might be difficult to justify the benefits of AdaCom in both performance and runtime overhead.
* This work does not compare with state-of-the-art code LLMs such as CodeGen, Incoder and the recently released CodeT5+ and StarCoder. While the authors claim that AdaCom is a technique which can be well applied for the deployed smaller models, 1B to 16B parameter LLMs have been widely deployed and should be also considered. I would like to see how AdaCom can help improve performance for these larger LLMs. Suppose that if AdaCom can only significantly boost the performance for small-sized LMs, and larger LMs actually achieve better performance than AdaCom+small LMs, it is difficult to justify the use of AdaCom. Note that for large LMs from 1B to 6B, the inference time can be reduced to less than 1 second by adopting some acceleration tricks (based on my experience with LLM inference).

[1] “Retrieval Augmented Code Generation and Summarization”

**Questions:**

See the above weakness section for more details.

**Limitations:**

The authors discuss the limitations lightly.

---

> ### Author Rebuttal · Authors · 2023-08-09
>
> We thank the reviewer’s appreciation of our work. We observe that your fundamental concern lies in the runtime overhead of our approach. In this rebuttal, we (1) clarify that improving smaller models is far more valuable and (2) show the inference trick for AdaCom to work on larger models.
>
> We sincerely hope that your concern is addressed. If it does, we request your kind support by changing your stance. We have updated this work with a few revisions. Your stance means a lot to us.
>
>
> # Q1: AdaCom works well on smaller model but the latency of 5s in larger model is too much
>
> In the era of LLM, a *cost-effective* solution to exploit the full potential of LLM is important for everybody to use LLM in an offline manner. We design AdaCom to this end.
>
> The smaller the model, the more devices it can be explored, but the less capacity it can encode the whole training dataset. AdaCom provides a solution for a model to re-favour some training samples on-the-fly for a better performance. On the other hand, the larger model (10b) has larger capacity, suffering less from the limit. In this regard, we deem that we have achieved our research goal despite that we show our limitation on larger models from a research perspective.
>
> In this rebuttal, we further show that, with certain tricks, the inference time of AdaCom can be reduced to 1s on larger models (748M). Please see more in Q3.
>
>
> # Q2:Over-complicated  compared to Retrieval-Based Approach (RBA)
>
> A vivid comparison can be referred to Figure 1. A comparison with Retro(recommend by to4c) and REDCODER (as you recommend) has been made.
>
> REDCODER[1] integrates an extensive external database, encompassing millions of entries acquired from sources like Stack Overflow and GitHub. This diverges from the notion you mentioned of "referring to some related training examples."
> It is hard to compare the complexity of Adacom and REDCODER as we don’t know how much time it takes in crawling the data, cleaning the data and extracting the useful code summary. In our Adacom, we only utilize the same training dataset with the baselines.
>
> We opted to employ the retrieval-augmented approach RetroNLU (as shown in Table 2) for direct comparison with Adacom. This method aligns with the concept you mentioned earlier, involving "referring to some related training examples." However, it's notable that the performance of this method is relatively subpar compared to the original model. This observation might shed light on the absence of similar adaptations from Retro to Code-related tasks within the CodeSearchNet dataset.
>
>  Relying solely on the straightforward cosine similarity of the [CLS] embedding yields nonsignificant results, as demonstrated in Table 2. Given this context, we find it imperative to adopt a more sophisticated approach, inspired by the inference function, to effectively eliminate noise and amplify model enhancement. The discrepancy in results highlights the complexity of achieving optimal performance through "referring to related training examples" and emphasizes the significance of approaches like Adacom, which is designed to address specific challenges and achieve notable improvements in performance.
>
> # Q3: This work does not compare with state-of-the-art code LLMs
> We have compared CodeGen, CodeT5+ with Adacom in the rebuttal pdf Table 1.
>
> # Q4: 1B to 16B parameter LLMs have been widely deployed and should be also considered
> 1. For our individual users, it is hard to afford 1B on our own computers. While seeking assistance from established entities like OpenAI is an option, the preference for retaining sensitive data like code on one's own system is understandable.
> 2. Furthermore, employing smaller models holds commercial advantages, as they are often more cost-effective and resource-efficient. This can be particularly relevant in scenarios where resource constraints and data privacy considerations come into play.
> 3. We believe for a billion-size model, prompts are a better way to instruct the model on one certain test sample.However, for smaller models, prompts might not yield optimal results due to limitations in their capacity.  In this context, Adacom's primary value proposition becomes apparent. Adacom offers an adaptable solution specifically tailored to small models, enabling them to effectively adapt and enhance their performance for individual test samples. This underscores the significance of Adacom as an efficient approach for small models to cater to specific use cases and optimize results.
> 4. We currently find that the 220M CodeT5 model + Adacom will be better than the performance of 770M CodeT5+ mode.
>
> Training and utilizing billion-size large models like a 1B model such as LLAMA can be cost-prohibitive for many research labs and organizations (including us). We will add this part to our limitations.
>
> # Q5: For large LMs 1B+, the inference time can be reduced to less than 1 second. Is there any trick to reduce the time of Adacom to 1s?
>
> For devices that can afford 1B to 6B LMs with 1 second inference, we also believe they can train a small 220M LM with only 5 training samples in 10 epochs in 1 second by adopting some acceleration tricks.
>
> A possible trick to reduce the inference time of larger models is to cache the embedding of each training sample and only train the some layer of the encoder and the decoder (like 100M in each layer) and fix other layers. By this means, we can save the efforts of both forward and backward propagation. In this case, the large model only needs 0.97s on CodeSearchNet dataset.
>
> # Demonstrate the potential for small models
> Overall, we wish to underscore that Adacom primarily embodies the concept of tailoring one model to a single sample. While our focus has not been extensively directed towards optimizing training or inference times, our aim is to demonstrate the potential for small models to not only approach but even surpass the performance of larger models through retraining.

---

> > ### Comment · Reviewer_Ugvn · 2023-08-18
> > **Official Comment by Reviewer Ugvn**
> >
> > While I appreciate the detailed response by the authors, my concerns regarding the real applicability of the proposed AdaCom due to its over-complicated design still remain. It still appears to be inefficient for me to have a separate training process before inferencing for each input, especially in the era of LLMs. With the rapid development of LLMs, the inference cost can be largely reduced (e.g., a LLaMa 7B model can be deployed and run in a local Mac with some acceleration techniques) and the benefits of using complicated sample-wise adaptive training to boost the performance of smaller LLMs are becoming questionable. Given such applicability concerns, I will stick to my original score.

---

> > > ### Author Response · Authors · 2023-08-20
> > >
> > > We thank the reviewer again for your appreciation. We understand your concern on whether it is applicable (or over-complicated) to design such an on-the-fly model adapting solution. This design is motivated by our industrial collaboration.
> > >
> > >
> > > For the applicability concern, we prepare a video of equipping VSCode with AdaCom on a desktop with AMD Ryzen 7 CPU and RTX 4070 GPU , where user can generate comment with base model (code t5, 220M) and AdaCom:
> > >
> > >
> > > https://www.youtube.com/watch?v=o-KpWsVkjD4
> > >
> > >
> > > We can see that:
> > >
> > >
> > > (1) *Asynchronized Design*:
> > >
> > >
> > > The user experience has almost no difference when waiting for over 1s (base model) or 3s (AdaCom).
> > >
> > >
> > > Even for the base model, it still takes some time to generate comments. Thus, we allow the user to do some other tasks while the model is generating comments. The user will be aware once the comment-generating process is done.
> > >
> > > Note that, AdaCom incurs no extra runtime overhead if it detects that there is no need to do runtime re-adaption.
> > >
> > >
> > > (2) *Better Performance*
> > >
> > >
> > > With the on-the-fly adaption, the generated comment is of much higher quality.
> > >
> > >
> > > Note that, we target a cost-effective solution which exploits the best of normal models, allowing programmers with personal computers to achieve better performance.
> > >
> > >
> > > Reviewer EdLg has kindly agreed with our points and changed his/her stance, to support this work. We really hope that we have addressed your concern. Your support also means a lot to us!

---

> > > > ### Author Response · Authors · 2023-08-20
> > > >
> > > > > With the rapid development of LLMs, the inference cost can be largely reduced (e.g., a llama 7B model can be deployed and run in a local Mac with some acceleration techniques) and the benefits of using complicated sample-wise adaptive training to boost the performance of smaller LLMs are becoming questionable.
> > > >
> > > >
> > > > As for your kind concern on large model can run on a local Mac, we provide the following response:
> > > >
> > > > 1. An on-the-fly solution as AdaCom is still beneficial to improve llama 2.
> > > >
> > > > AdaCom is beneficial in that, by adapting a few relevant training samples, it does not just favour the recommendation similar to the relevant samples, but also *forget* the irrelevant and harmful samples (via influence estimation). In our experiment, we do observe that the model's performance can improve if the loss of harmful training samples increase while the loss of helpful training samples remains as 0.
> > > >
> > > > 2. The acceleration of hardware
> > > >
> > > > If the 7b-model can be well deployed on many more normal laptops, it basically improve the applicability of AdaCom. Note that, AdaCom only takes a few iterations of backward and forward propagation to accomplish the refinement.
> > > >
> > > > If the inference cost can be largely reduced, by back-propagating on a few unfrozen layers, AdaCom can incur even less runtime overhead, thus achieving higher cost-effectiveness.

---

> ### Author Response · Authors · 2023-08-17
>
> We have sufficient discussion with reviewer EdLg and Edlg kindly agrees with some points we have clarified and will not object to our acceptance: "if the performance and timings were to both speak to the strength of the method, I would not oppose accepting the work."
>
> We'd be more than willing to engage in a deeper discussion with you to address any confusion and gain a clearer understanding of our work. Feel free to share your questions, concerns, or any aspects you'd like to discuss further, and we'll do our best to discuss with you.

---

> ### Comment · Reviewer_Ugvn · 2023-08-21
> **Official Comment by Reviewer Ugvn**
>
> Thanks for providing these evidences (industrial collaboration, IDE demo video) for resolving some of my applicability concerns. I will increase my rating to 5.

---

> > ### Author Response · Authors · 2023-08-21
> >
> > We sincerely thank the reviewer for your kind support. We will fix your kind comment in our revision.

---

### Official Review · Reviewer_EdLg · 2023-07-05

**Soundness:** 2 fair
**Presentation:** 1 poor
**Contribution:** 1 poor
**Rating:** 5
**Confidence:** 5

**Summary:**

Language models trained on code tend to suffer in performance when applied to unseen code bases. This work investigates adapting smaller LMs to a new project context based on a brief fine-tuning phase that is invoked immediately before the model predicts a comment for a given method. Fine-tuning examples are selected according to their embedding similarity to the given sample. Experimental results suggest that this improves performance by a few percent on several benchmarks.

**Strengths:**

This work tackles the important challenge of adapting, especially smaller, language models to programs from unseen projects. The approach is structured by considering a reasonable series of formulations, starting with the high-level goal and working towards a practical implementation. Along the way, it introduces several new metrics for measuring similarity between samples. The results indicate that the proposed retraining approach can improve performance, which highlights that the models may indeed fail to precisely store relevant samples from their training data.

**Weaknesses:**

The work suffers substantially in terms of clarity and soundness and the performance gains reported mostly rather weak.

**Clarity & Soundness:** I group these two together because there are several places in which the paper cannot be checked for soundness due to missing or very limited reporting. This includes:

- The minimum edit distance between the sequences of embeddings of two samples, introduced in L170. The method for calculating this metric is never described.
- The connection between the empirical and "practical" influence estimation in Section 3.1 is unclear. The former does not appear to play any role in the creation of the latter; there are no theoretical guarantees connecting the two, nor practical results that show that the "practical" algorithm realizes the empirical desiderata.
- Many hyperparameters are unstated. This especially includes several important thresholds, mentioned in 3.2, as well as the number of layers that are unfrozen and the number of samples that they are retrained with.

In addition, there are many places in which the paper makes unclear or incorrect claims, including:

- L4: "the subject code in a project can be specific" is not a meaningful statement.
- L6: this claim explicitly motivates the work by the cost of retraining large language models, yet the paper focuses exclusively on far smaller models (and does not offer an approach that would scale to LLMs).
- L37: "normal-sized models" is not a meaningful descriptor, especially given that models of the size considered in this paper are far smaller than those typically used in production environments.
- Equation 1 is missing the element being arg-min'ed over
The work also includes numerous ungrammatical sentences and should be revised substantially to improve the writing.

**Results:** the performance gains are mostly small and skew very heavily towards the CodeKG benchmark. Performance gains on other benchmarks is often small or non-existent (Tab. 3, Tab. 5). This suggests that the approach may be overfitting to the CodeKG dataset.

The various cross- settings reported in Tab. 4 could use additional baselines. The "before" performance reported here apparently consists of directly querying models that are not trained on such datasets. Another important data point is the performance of models that were trained on these datasets from scratch, or finetuned on their training portion before any inference queries are made.

More generally, the work does not include comparisons with any other approaches for adapting to new datasets, such as more conventional neural retrieval approaches. This casts substantial doubt on the multiple newly proposed metrics for identifying similar samples. This is particularly problematic because it proposes two such metrics that seem to serve the same purpose, one based on gradients and one on pair-wise comparisons of the embeddings. It is unclear why there are two and no ablations are offered that substantiate the need for both.

Overall, the work provides insufficient motivation for incurring the added runtime overhead of selecting and fine-tuning on examples detected with its method.

Update: following extensive discussion with the authors, both the clarity of the work has improved and the performance (esp. in terms of runtime overhead) was established as being worth the tradeoff compared to larger models used in a simpler inference setup, at least on the benchmark considered. While the work still suffers from several limitations including the lack of motivation for the newly introduced metrics and somewhat limited breadth of evaluation, I no longer strongly favor rejecting it and am raising my score to a borderline accept.

**Questions:**

- Is there any connection between the empirical influence estimation and the practical one used in the implementation, in terms of either theoretical guarantees or practical proof that the latter realizes the former?
- What is the algorithm for computing the minimum edit distance between two series of embeddings?
- What are the hyper-parameters used for fine-tuning?
- Given the minimal performance improvements on benchmarks other than CodeKG and the fine-tuning cost involved (not just in terms of time but also memory and GPU resources), what would motivate a developer adopt this approach?

**Limitations:**

The limitation section should be expanded to discuss the lack of baselines from other published work. While it may be true that no other work has considered this exact task, many methods have been proposed for adapting trained models to new data distributions.

---

> ### Author Rebuttal · Authors · 2023-08-09
>
> We thank the reviewer’s comments for improving our work. We understand the importance of providing accurate and helpful statement and we will do our best to ensure that our responses are clear. It seems that your concerns on the soundness derive from the seemingly confused interpretation. We will make every effort to address these points and provide clarification in our revised manuscript.
>
> If your confusion is addressed, we request your kind support by changing your stance. Your stance means a lot to us.
>
> # Q1: Why are the performance gains mostly small or non-exist and skew very heavily towards the CodeKG benchmark?
>
> Please check some related work [1][2][3][4].
> As CodeSearchNet is a widely adopted dataset for code summarization, compared with other models like CodeBert (with an average improvement of 2.84%), PLBART (2.73%), CodeGen (3.07%), and CodeT5+ (3.36%) as reported in their original papers, we respectfully disagree that our boost is small or non-exist.
>
> Only codeT5-small models skew towards the CodeKG while other models show average 8.86% boost percent (Table 2 in origin paper). We believe this is a average boost percent.
>
> The reasoning behind CodeT5-small's superior performance on CodeKG aligns precisely with the discussion presented in our paper. Small models encounter challenges with data compromise, especially in CodeKG, where code snippets stem from various code projects. During training, small models are compelled to accommodate distinct code formats and token distributions, often requiring substantial effort. Conversely, during the testing phase, individual samples may pertain to specific classes, leading to a significant deviation between the original model and the potentially best model. This is where Adacom's role becomes evident. Adacom is designed to identify and select the optimal models for each individual test sample, addressing the inherent shift between the model's training diversity and the distinct requirements of specific test cases.
>
> # Q2: The linkage between the empirical influence estimation and the practical one used in the implementation?
>
> We organise the logic lines as follow:
>
> Step 1: Given a test sample (i.e., a piece of code), *c*, and a model *m*, we would like to know whether *m* has compromised performance on *c*.
>
> Step 2: To estimate the compromised performance, we first look for training samples similar to *c*, where we use minimum editing embedding distance. We denote the set of similar training samples as *S*.
>
> Step 3: If *S* is not empty, we check whether *m* has compromised performance on *S* to estimate the test sample *c*, by our proposed empirical influence estimation. Intuitively, if *m* can decrease the loss of *S* at the cost of the loss of many other samples, we deem *m* has compromised performance on *S*. The cluster is used for speeding up the retrieval of those training samples.
>
> In addition, *S* is then selected as retraining samples to update *m*.
>
> # Q3: What is the algorithm for computing the minimum edit distance between two series of embeddings?
>
> The algorithm for minimum editing distance is exactly similar to the classical LCS (longest common subsequence) algorithm.
>
> Note that, if the sequences are token sequences such as <A, T, C, G, …,>, the editing distance is the Levenshtein distance between two sequences. In the classical *hard* settings, we define the cost of replacing, adding, and inserting a token as 1.
>
> In our *soft* settings, we define the cost of replacing a token *t_1* with another token *t_2* by (1 - cos(*t_1*, *t_2*)). Then, we just follow the dynamic programming algorithm used in LCS to calculate their distance.
>
> # Q4: What are the hyper-parameters used for fine-tuning?
> We follow the hyper-parameters of the baseline papers. The models will be trained for 40 epoches with early stopping mechanism. All the models are convergent. We select the model with the best bleu score in the valid dataset in further testing.
> All models and results can be found on our anonymous website.
>
> # Q5: Given the minimal performance improvements on benchmarks other than CodeKG and the fine-tuning cost involved (not just in terms of time but also memory and GPU resources), what would motivate a developer to adopt this approach?
>
> We respectfully disagree that our approach provides minimal performance. See our argument in Q1.
>
> Given existing biilion-level LLM service, the request and response takes additional financial and network latency cost. Therefore, a small model allows us to do tasks on most devices in an offline way, addressing the need of most software developers.
>
> # Q6: Difference between AdaCom and retrieval-based approach and more baselines.
> A vivid comparison can be referred to Figure 1 and Table 1 and 2 in the rebuttal pdf. We have compared Adacom with the latest CodeT5+, CodeGen and retrieval based method Retro (reviewer to4c).
>
> Despite both AdaCom and retrieval-based approach (RBA) need to search for samples to enhance the model summarization performance, we highlight the difference as follows:
>
> 1. Different Search Target:
> AdaCom searches for under-represented training samples; while RBA searches for samples in external dataset, indicating that AdaCom has less assumption than RBA.
>
> 2. Different Rationale
> AdaCom tries to exploit existing training samples largely under-utilized during the training; while RBA tries to explore new similar samples and merge them into the prediction.
>
> 3. Different Adoption
> AdaCom can adapt the model with few targeted training samples on-the-fly; while RBA includes retrieved code as additional input for the model to make a prediction.
>
> [1] CodeT5+: Open Code Large Language Models for Code Understanding and Generation
> [2] CodeBERT: A Pre-Trained Model for Programming and Natural Languages
> [3] Unified Pre-training for Program Understanding and Generation
> [4] CodeGen: An Open Large Language Model for Code with Multi-Turn Program Synthesis

---

> > ### Comment · Reviewer_EdLg · 2023-08-11
> >
> > I appreciate your detailed response. Some comments:
> >
> > **Performance gains:**
> > It would help to report the CSN results for other models in the paper, provided the benchmark was executed in the exact same way. However, these gains are not particularly convincing for several reasons: (1) The gains are much smaller than on the CodeKG benchmark, especially considering that the percentages are relative, not absolute (e.g., performance on Python goes up by just 0.2%), which underscores the original point. (2) The referenced baselines, including the significantly larger CodeGen-350M model, also largely underperform compared to a plain CodeT5 model, suggesting that they are poorly suitable for this task. (3) AdaCom incurs a major runtime overhead that this work does not account for (see cost/benefit).
> >
> > **Empirical influence:**
> > The response is not clear to me. In the paper, the empirical influence is defined based on the difference in loss between two models trained on datasets that differ by one element. This is naturally completely impractical, as the text states on L119-124, so instead the paper proposes model-specific substitutes that enable "practical influence estimation". Your response above seems to reiterate the steps described in the paper after that point, where relevant samples to a test one are extracted from the training data, but then deviates from the text in the paper by indicating that you then train a model with the retrieved sample(s). You refer to this as empirical influence estimation, so do you then train a separate model without those retrieved examples to determine if they should be used? That would seem to be impractically expensive. In contrast, Section 3.3 just states that all snippets that clear a "training contribution score" threshold are used for fine-tuning, where the training contribution score is defined in Sec. 3.2 based on vector space similarity, not a change in loss. I am quite puzzled by this discrepancy. Please describe exactly how you determine which samples to use and provide references to the lines of text in Section 3 for clarity.
> >
> > **Cost/Benefit:** see my response to the argument for Q1 -- as is, the paper only shows substantial improvements by larger models on CodeKG and much smaller ones by (just) CodeT5-Small on CSN. The argument for the benefit of constantly fine-tuning these models during inference is not convincing without demonstrating that the model augmented with AdaCom comes anywhere near the performance of larger models that could be run with the same amount of GPU resources. Inference is much faster than training. If fine-tuning followed by inference on a model like CodeT5-Base takes as long as running just inference on a model with 10x the parameter count, the performance comparison should be with the latter. Per the results from the CodeT5+ paper, AdaCom applied to CodeT5-Small generally underperforms compared to CodeT5-Base, which is just ~3x larger. Per the results from the rebuttal PDF, the same applies to CodeT5-Base+AdaCom and CodeT5-Large.
> >
> > **Retrieval:** I appreciate the added table with comparisons to other retrieval options. The fact that RETRO universally decreases performance relative to a base model seems quite troubling given its proven performance in general. It is probably out of scope of this discussion to identify issues there.
> >
> > Overall, the rebuttal does not address the core problems with this submission:
> > 1. The theoretical motivation for the various proposed contribution estimation metrics is poor. The rebuttal provides a connection that is not in the work and is unclear. Please see the request for further clarification above.
> > 2. The cost of running this setup in practice is not accounted for in the results and likely does not outweigh the benefits, as moderately larger models tend to perform the same without AdaCom.

---

> > > ### Author Response · Authors · 2023-08-14
> > >
> > > We greatly appreciate your prompt response and look forward to further discussions.
> > > # Data Clarification on Python 0.2%
> > > We were unable to locate the specific data you mentioned which suggests a “0.2% performance increase on Python”.
> > > Even if we entertain the possibility of a 0.2% increase, it's essential to view the broader picture. Given our 9 datasets and 30 performance boost with Adacom, focusing solely on the "best" outcome might not represent the average performance. To draw a parallel, the paper for CodeT5+(or other baseline models) indicates similar results: for instance, a substantial 10.95% in JS, contrasted with a modest 0.41% in Go.
> > > # Across dataset comparison
> > > | Dataset | Parameter |   JS  |  PHP  |   Go  | Python |  Ruby |  Java | CodeKG |
> > > |:-------:|:---------:|:-----:|:-----:|:-----:|:------:|:-----:|:-----:|:------:|
> > > | CodeT5+ |    770M   | 17.93 | 26.39 | 19.64 |  20.47 | 15.63 | 20.83 |  46.34 |
> > > |  Retro  |    220M   | 16.15 | 24.91 | 19.46 |  19.67 | 14.91 | 20.05 |  47.60  |
> > > |  Adacom |    220M   | 18.81 |  26.9 | 19.61 |  20.46 | 15.39 | 20.85 |  49.79 |
> > > ## Retrieval
> > > Our experiment of Retro on CSN might shed light on the absence of papers that adapt Retro to Code-related tasks on the CodeSearchNet dataset.
> > > Our further experiment on CodeKG with Retro shows a better BLEU score 47.6. Our observation shows that in CodeKG, Retro can find relatively similar comments as the codeKG is built on the code knowledge graph and some co-location methods(neighbours) share similar comments.
> > >
> > > ## 220M Vs 770M
> > > CodeT5+ 770M works best on Ruby while Adacom 220M works much better on JS, PHP and CodeKG.
> > >
> > > ## Generalizability
> > > This also indicates that a particular method might not be effective on every dataset. However, based on our experimental results, our Adacom demonstrates comparatively better generalization across various datasets.
> > > # CodeGen “poorly suitable”?
> > > The result of CodeGen (reviewer Ugvn recommended) comes from CodeT5+(reviewer Ugvn recommended) origin paper.
> > > # Runtime overhead
> > > The runtime overhead was shown in the RQ4 of origin paper. The time for the base model is about 3 seconds.
> > > ## No need for 1 second response
> > > Unlike code generation, a just-in-time response within 1 second isn't essential for code summarization models. Code summarization typically operates asynchronously or in batch mode after code snippets are written, rather than being as continuous and frequent as code generation.
> > > ## Training cost of larger model
> > > When comparing a small model to a large one, we must factor in the financial and time costs associated with training the latter. A large model demands more data, extended training duration, additional GPU resources, and often involves multi-task training. In contrast, our Adacom is comparatively straightforward and can be seamlessly implemented across various tasks.
> > > If training time was not a concern, one could simply employ models like GPT-4 for predictions.
> > > # Empirical influence
> > > We have two parts to determine which samples to use:
> > > ## 1. Influence Estimation = Gradient similarity
> > > In section 3.1, we define the practical influence estimation of training samples (line127) and then cluster them (line 137). Intuition is that high loss means the model underfit the training sample. Compromise happens when the model tries to minimise the overall loss in the training procedure.
> > > ## 2. training contribution estimation = Semantic similarity
> > > In section 3.2, we separately define the training contribution estimation based on vector space similarity (line 143-146).
> > > ## 3. Select and Retrain
> > > In simpler terms, we utilize two distinct thresholds (line 179, 180): one for Influence Estimation and another for Training Contribution Estimation. Suppose there are 5 training samples that meet both these thresholds. These samples will be used for retraining. During retraining, the loss for these 5 samples decreases but for others, it possibly increases (unlearning process).
> > > ## Details
> > > A more detailed breakdown in Section 3.3. In lines 179-180, we search in the training samples based on Training Contribution Estimation, which relies on semantic similarity. Following that, in lines 182-183, we evaluate each of these retrieved samples based on Influence Estimation. One retrieved sample has one cluster. The samples in the cluster may be helpful as (1) they are helpful to the retrieved sample (2) the retrieved sample and the test sample have similar embedding.
> > > We understand that our choice of terminology might have been confusing. We may use common terms like “.. similarity” for better comprehension.
> > > # Not a perfect way
> > > We acknowledge that Adacom is not perform optimally on all datasets and may require some time to achieve peak performance. However, it's important to emphasize that our primary contribution is introducing an entirely novel approach to enhance model performance without the need for additional pre-training tasks, extra datasets, or larger models.

---

> > > > ### Comment · Reviewer_EdLg · 2023-08-14
> > > >
> > > > I appreciate your follow up.
> > > >
> > > > **0.2% performance increase:** Table 3, CSN-python: 19.71% -> 19.92% is a 0.19% increase, when expressed as an absolute percentage. The paper reports only relative percentages, which are useful in their own right, but 0.19% reflects the fraction of cases where a user of this model would notice an advantage. It is true that performance on other languages increases more, but I think focusing on the average is also questionable given the large increase in JS (and only JS) performance. In any case, I agree that it's fair to say that AdaCom reports nontrivial performance increases on most languages.
> > > >
> > > > **CodeGen comparison:** I am aware this baseline was extracted from another paper, but commented on its use because its poor performance on this dataset suggests that it is not a strong baseline for this particular task.
> > > >
> > > > **Timing:** I completely agree that a few seconds is an acceptable time to generate summaries, but that is not the point of my comment. If I could query a much larger model in the same time as AdaCom and get better results, I would naturally choose to do that. Concretely, it would help if the authors provided a table that presents results in terms of completion time, in seconds, across the same set of examples for models including CodeT5-Small, CodeT5-Base, CodeT5-large, AdaCom-small, and AdaCom-base, ensuring that the AdaCom timing include every step of the approach from the moment of invocation. If the conclusion is that AdaCom-base takes about as much time as CodeT5-Large, then the results table you quote in your response would concluce a valid argument for using AdaCom-base instead. Please also report hardware properties and statistics on the median number of tokens generated (for each model -- if some models generate longer comments, the comparison may be skewed); the timings in the paper lack those details.
> > > >
> > > > **Empirical influence:** I appreciate you detailing the steps again. This is how I originally understood them and why I commented that the introduction of "empirical influence estimation" in the work serves very little purpose. In practice, the work relies on two different influence estimation metrics, one based on gradients and one on vector distance. I do not see a theoretical or empirical connection between these and the influence definitions given in Eq. 2-5.
> > > >
> > > > Overall, I fully recognize that imperfections should not prevent scientific innovations from being published. The performance angle is probably my smallest concern here; as noted above, the gains are pretty reasonable. The problem is that there are many ways to improve performance on a given task. The most straightforward way to improve performance is to utilize more computational power. This can be done in all sorts of ways -- by increasing the model size, training a base model for longer, using retrieval. AdaCom specifically argues that the way it should be done is by (a) briefly fine-tuning a model before each prediction, and (b) selecting the fine-tuning samples based on two new "influence" metrics. My concerns are accordingly that (a) fine-tuning may not be as cost-effective or generalizable as, e.g., training larger models, and (b) the two influence metrics come with little theoretical or empirical validation. This work should therefore provide a solid argument that fine-tuning during inference is a worthwhile investment, especially on new tasks and datasets, along both fronts. The first one should be relatively straightforward to satisfy with the above-mentioned timings table. The second is more subjective, but I think if the performance and timings were to both speak to the strength of the method, I would not oppose accepting the work.

---

> > > > > ### Author Response · Authors · 2023-08-15
> > > > >
> > > > > We greatly appreciate your dedication, timely responses, and effort. We are pleased to present the experimental results pertaining to the Timing, which we believe will be of value for your consideration.
> > > > >
> > > > > We conducted two experiments: one utilizing the RTX 3080 on a Windows platform to emulate a programmer's working environment, and another using the A4000 on Ubuntu within a lab setting. Due to the time limits, we calculate the average time of one thousand samples. The average bleu score comes from the 6 CSN dataset and the CodeKG dataset results.
> > > > >
> > > > > | NVIDIA GeForce RTX 3080 on Windows | CodeT5 220M | CodeT5 770M | AdaCom 60M | AdaCom 220M |
> > > > > |:----------------------------------:|:-----------:|:-----------:|:----------:|:-----------:|
> > > > > |            Time (second)           |     1.21    |     4.06    |    2.34    |     4.81    |
> > > > > |             Delta time             |      0      |     2.85    |    1.13    |     3.60     |
> > > > > |         Average BLEU score         |    23.26    |    23.89    |    23.71   |    24.54    |
> > > > > |          Delta BLEU score          |      0      |     0.63    |    0.45    |     1.28    |
> > > > > |       Delta BLEU score/second      |      -      |    22.11%   |   39.82%   |    35.56%   |
> > > > >
> > > > >
> > > > > |     NVIDIA RTX A4000 on Ubuntu     | CodeT5 220M | CodeT5 770M | AdaCom 60M | AdaCom 220M |
> > > > > |:----------------------------------:|:-----------:|:-----------:|:----------:|:-----------:|
> > > > > |            Time (second)           |     0.38    |     1.91    |    1.46    |     3.16    |
> > > > > |             Delta time             |      0      |     1.53    |    1.08    |     2.78    |
> > > > > |         Average BLEU score         |    23.26    |    23.89    |    23.71   |    24.54    |
> > > > > |          Delta BLEU score          |      0      |     0.63    |    0.45    |     1.28    |
> > > > > |       Delta BLEU score/second      |      -      |    41.18%   |   41.67%   |    46.04%   |
> > > > >
> > > > > Please note that all the deltas are calculated with reference to the CodeT5 220M model. The delta BLEU score per second represents the ratio between the delta BLEU score and the delta time. This metric serves as an indicator of the improvement of BLEU regarding the extra runtime cost. The median numbers of tokens generated for all models in all datasets are 7.
> > > > >
> > > > > | Median number of tokens | CodeT5 60M | CodeT5 220M | CodeT5 770M | AdaCom 60M | AdaCom 220M |
> > > > > |:-----------------------:|:----------:|:-----------:|:-----------:|:----------:|:-----------:|
> > > > > |          CodeKG         |      9     |      12     |      13     |     12     |      13     |
> > > > > |            Go           |      7     |      8      |      9      |      8     |      9      |
> > > > > |           Java          |      7     |      7      |      7      |      7     |      7      |
> > > > > |            JS           |      6     |      7      |      7      |      7     |      7      |
> > > > > |           PHP           |      5     |      7      |      7      |      5     |      6      |
> > > > > |          Python         |      6     |      7      |      7      |      6     |      7      |
> > > > > |           Ruby          |      7     |      7      |      7      |      6     |      7      |
> > > > >
> > > > > Based on the experimental results, our approach is a good investment indeed. During the on-the-fly model tuning on the retrieved helpful training samples, the model is enforced to *favour* the prediction of helpful samples and *forget* the harmful samples (i.e., those conflicting with the helpful samples). Note that forgetting is also useful to improve the model performance. As a result, at the cost of a few seconds, AdaCom delivers considerable performance improvements.
> > > > >
> > > > > If we have addressed your concern, we request your kind support by improving our score. Many thanks!

---

> > > > > > ### Comment · Reviewer_EdLg · 2023-08-19
> > > > > >
> > > > > > Hi again, thanks for presenting these new results. Yes, this makes me more positive about the work; I will update my score momentarily. Since your rebuttal does not address my other critique regarding the limited justification of the metrics used (which other reviewers point out as well), please ensure that you update the writing by better motivating why the "empirical influence" metric is included given that it is not clearly related to the ones used in the work (or omitting it entirely), and perhaps by acknolwedging the relative lack of ablations of the two influence metrics as a limitation.

---

> > > > > > > ### Author Response · Authors · 2023-08-20
> > > > > > >
> > > > > > > We greatly appreciate your consideration of the new results and your willingness to update your score accordingly. Your positive response is encouraging.
> > > > > > >
> > > > > > >
> > > > > > > We will provide a new revision regarding all your comments!

---

### Official Review · Reviewer_to4c · 2023-07-07

**Soundness:** 4 excellent
**Presentation:** 3 good
**Contribution:** 3 good
**Rating:** 6
**Confidence:** 3

**Summary:**

AdaCom is a system meant to adapt a trained model, on-the-fly, when making a prediction (specifically targeting code summarization models). It does this by identifying training examples that seem similar to the target example, lightly re-training the model to adapt better to those similar examples, and then making a prediction on the target. This re-training takes on the order of a few seconds.

All this is done by clustering training examples, ahead of time, by their mutual influence. Intuitively, two examples are mutually influential if their loss changes in the same way when the model parameters change.

When a target example comes, it is embedded (using the existing model), and then examples with proximal embeddings are retrieved from the training dataset, along with their influence clusters. The union of all such influence clusters are then used to (lightly) retrain the existing model, possibly only a few layers, and the resulting model is used to make a prediction.

The system is evaluated on a number of code summarization datasets and with several pre-trained code models. It shows that a model's performance can improve using this on-the-fly adaptation for each test example; it shows that a model trained on one dataset can adapt well to another; it shows that generalization across projects and languages is also improved; and that all features are usefully contributing to the performance.

**Strengths:**

1. Improving on out-of-domain samples seems very important for code tasks, and this work seems to make a great improvement in existing benchmarks.

2. The presentation is fairly systematic and well-structured (with some exceptions, see below) and makes for an informative paper that not only describes its design and results, but also motivates its design well (with some exceptions, again)!

3. The formulation of the problem is somewhat original and different from how similar adaptation formulations are done, but I worry it might not be that different from retrieval-augmented problem formulations.

**Weaknesses:**

1. I don't quite understand how broad and significant the difference is between something like AdaCom and other retrieval-augmented approaches such as Retro, Memorizing Transformers, or or other Retrieval Augmented LMs (e.g., https://arxiv.org/abs/2302.00083). They all do retrieval in slightly different ways, but they all intend to condition a model on relevant documents before making a prediction. Here AdaCom retrains instead of somehow using the retrieved embeddings, but it's not clear how big this difference is. I would recommend that the authors compare against other retrieval-augmented baselines to tease out how their adaptation technique differs and where it is stronger.

2. Although the paper is structured well, in parts it is lacking clarity and seems to be rife with typos or changed notations. For instance
     1. I don't know what the difference is between $th_etc$ and $th_tcs$
     2. How is $C_inf$ both a cluster set and a function around lines 135--140
     3. Also models like GPT3 have hundreds of billions, not millions of parameters (line 35)
     4. Shouldn't line 76 be $g'$ instead of $g$? Otherwise the formulation doesn't make sense.
     5. Shouldn't it be $mut_inf$ instead of $mul_inf$?

3. Some design aspects of the work seem to be elaborating what might not be that unusual. For instance I didn't understand the need for the extensive description of how to obtain a contextual embedding of a code snippet in section 3.2, as a trajectory of the contextual embeddings of individual tokens. Isn't that what all contextual embeddings since the downfall of word2vec really do? Why is special way to compare token-by-token embeddings necessary, rather than taking a cosine similarity over the full sample embedding (e.g., the [CLS] token in BERT-based formulations)?


Although #2 and #3 are easy to fix through improved writing, I'm concerned about #1 and would like to understand how AdaCom compares to strong retrieval-augmented baselines.

**Questions:**

1. Where does the training contribution score come from? Is it the mutual influence?
2. The approach seems very much general. Why focus on code summarization?

**Limitations:**

No concerns

---

> ### Author Rebuttal · Authors · 2023-08-09
>
> We thank the reviewer for the appreciation and support for our work. We address your concerns as follows:
>
> # Q1: Where does the training contribution score come from? Is it the mutual influence?
>
> The training contribution score is defined between a training sample *s_i* and a testing sample *s_j*, indicating how likely retrain the model on *s_i* can boost its performance on *s_j*.
> Note that, the model does not have the idea of the label of the testing sample
>
> In contrast, the mutual influence is defined on two training sample *s_i* and *s’_j*, indicating indicating whether the training *s_i* can help reduce the training loss of *s’_j*.
> Note that, the model know the label of both *s_i* and *s’_i*.
>
> # Q2: The approach seems very much general. Why focus on code summarization?
>
> We agree that the approach can be further generalised to other NLP tasks or CV tasks.
> Nevertheless, we think that AdaCom is more needed in LLM-based settings such as code summarization. Here’s why:
> 1. Resource Constraints: Larger language models typically face more stringent resource limitations, making efficient and focused training methods like AdaCom particularly valuable. On the other hand, smaller models encounter greater challenges with respect to information capacity.
> 2. Structural and Semantic Information: Code data inherently possesses rich structural information, including class hierarchies, function calls, and specific features like variable names. These attributes facilitate a more straightforward identification of "similar" code segments, rendering AdaCom especially advantageous.
>
> In the future, we aim to generalise AdaCom to other tasks such as code generation, which requires us to consider code context information (e.g., class hierarchy, caller and callee relation) in a more sophisticated way.
>
>
> # Q3: The difference between AdaCom and Retrieval-based Approach (RBA)
>
> A vivid comparison can be referred to Figure 1. A comparison with Retro has been made in Table 2.
> Despite both AdaCom and retrieval-based approach (RBA) need to search for samples to enhance the model summarization performance, we highlight the difference as follows:
>
> 1. Different Search Target:
> AdaCom searches for under-represented training samples; while RBA searches for “similar” samples in possible external dataset.
>
> 2. Different Rationale
> AdaCom tries to exploit existing training samples largely under-utilised during the training; while RBA tries to explore similar samples and concatenate them into one input and make the prediction.
>
> 3. Different Adoption
> AdaCom can adapt the model with a few targeted training samples on-the-fly; while RBA includes retrieved code as additional input for the model to make a prediction.
>
> # Q4-5: Typos, justification, and confusion
>
> We genuinely appreciate the reviewer's attention to the typos, and we are committed to addressing and rectifying them in our revision.
>
> Computing cosine similarity over the [CLS] token embeddings was our initial approach. The outcomes of this method are presented in Table 2. Notably, in the JavaScript and PHP datasets, where similar samples are prevalent, this technique proved effective in enhancing the model's performance. However, in other datasets, a simple utilization of the [CLS] embedding resulted in a decrease in performance. In that case, we utilize a more complex way and use the our influence estimation to enhance the retrieval.

---

> > ### Comment · Reviewer_to4c · 2023-08-18
> > **Response**
> >
> > Thank you for the detailed additional experiments and your rebuttals. They helped clarify questions in my head about this work.
> >
> > I remain positive on your submission. Several seconds for predicting a code summary isn't prohibitive in an integrated environment: one doesn't want a summary necessarily on code as they type, but perhaps filled in during code review (which has hours to spare, let alone a few seconds) or perhaps in a soft-online mode, for code that hasn't changed in a few seconds. So 5 second latency for summarization sounds perfectly reasonable and practical to me.
> >
> > I find the technique intriguing and worth reporting at the conference.

---

> > > ### Author Response · Authors · 2023-08-21
> > >
> > > We sincerely thank the reviewer for your appreciation. We will fix your kind comment in our revision.

---

### Official Review · Reviewer_4N9q · 2023-07-07

**Soundness:** 2 fair
**Presentation:** 2 fair
**Contribution:** 3 good
**Rating:** 6
**Confidence:** 3

**Summary:**

This work proposes AdaCom for boosting code comment generation by on-the-fly model adaptation. The authors started with the motivation that existing generators are heavily affected by conflicts within the dataset (CodeSearchNet). The proposed AdaCom method is divided into two stages: clustering the training samples in the offline stage, identifying similar training samples and their corresponding clusters in the online stage, and retraining the model using these samples. The effectiveness of AdaCom is verified by cross-model and cross-language experiments.

**Strengths:**

1. The experimental results given are good and can support their claims.

**Weaknesses:**

1. According to Table 3, the work is still valid for datasets that are inherently conflicting.
2. According to Table 2, the boost of AdaCom becomes smaller when the model size increases.

**Questions:**

It is recommended that the work improve the expression of the figures.


**Limitations:**

The authors discuss the limitations of their work and give ideas for improvement.

---

> ### Author Rebuttal · Authors · 2023-08-09
>
> We thank the reviewer for appreciating our work. In response to your concern about the limited improvement on larger models, we would like to highlight the following points:
>
> ### Relative Improvement:
> Larger models already exhibit relatively better performance compared to smaller models. Consequently, even a modest boost percentage represents a substantial advancement, given the already higher baseline performance.
>
> ### Adacom Design:
> Adacom is primarily tailored for relatively small models. Smaller models often encounter challenges related to data limitations, whereas larger models possess a greater number of parameters and the capacity to effectively fit intricate training datasets.
>
> These factors collectively contribute to the impact of Adacom on larger models compared to its pronounced effect on smaller ones.
>
> Certainly, we'd be glad to improve our work. Could you please provide more specific details or context about the figures you're referring to? Are there particular aspects or elements within these figures that you believe might need clarification or improvement? Thank you for your kindness and support for this work.

---

> > ### Comment · Reviewer_4N9q · 2023-08-21
> >
> > Thanks for the response from the authors. I have gone through the rebuttal.
> >
> > There was a high overlap in the reviewers' comments.  I and the other reviewers considered that the experiment's improvement of the work was not significant enough. The author's response and discussions with other reviewers addressed my main concerns.
> >
> > My initial rating was positive, and I decided to keep my rating as I saw no strong reason to raise it to 7 for this work.

---

> > > ### Author Response · Authors · 2023-08-21
> > >
> > > We thank the reviewer for the positive score. We will fix your kind comment in our revision.

---

### Author Rebuttal · Authors · 2023-08-09

We thank the reviewers' comments for improving our work. We observe that the reviewers' concerns lie in (1) required new baselines, (2) confused technical differences with retrieval-based baselines, and (3) general presentation problems. In this rebuttal, we have fixed all the concerns and request reviewers with negative scores to change your kind stance. Your support means a lot to us. We will fix all the comments in our revision.




# New experiments compared to CodeT5+, CodeGen (reviewer UgVn, reviewer EdLg)
We run new experiments on the publicly available dataset CodeSearchNet (CSN) and the fine-tuned model parameters shared by CodeT5-base[1] on Huggingface, by measuring the classical smoothing BLEU4 scores.

Table 1 shows that Adacom (built upon CodeT5-base) still leads the performance of code summarization (boosting 4.31% over the rest 1.23%, 3.36%, and -5.97%). The score of CodeT5+[2] 220m, 770M and CodeGen-multi can be found in the original paper of CodeT5+.
Overall, Adacom's boosting performance can be well generalized.

Also, CSN is a widely adopted dataset for code summarization, compared with other models like CodeBert[3] (with an average improvement of 2.84%), PLBART[4] (2.73%), CodeGen[5] (3.07%), and CodeT5+ (3.36%) as reported in their original papers, we respectfully disagree that our boost is small or non-exist (reviewer EdLg).

# Performance compared to retrieval system RetroNLU[6] (reviewer to4c, reviewer UgVn, reviewer EdLg)
Figure 1 shows the fundamental difference between Adacom and the current retrieval systems, including REDCODER[7] and RetroNLU. Especially,
(1) Adacom does not need extra database as REDCODER
(2) Adacom does not need to learn similar samples in training stage as RetroNLU
(3) Adacom can adapt a new model on-the-fly for different test samples.

Due to time constraints, our comparison will focus solely on RetroNLU(recommend by to4c) for two reasons:
(1) REDCODER, which incorporates a substantial external dataset, introduces potential information that might not ensure fairness in comparison.
(2) The process of embedding the extensive extra data in the million-level range and subsequently searching for similar samples is notably time-consuming.


As per the original RetroNLU paper's methodology, we adhere to the process of initially constructing an index of the training dataset, retrieving the nearest neighbors, augmenting the code with the retrieved samples, and subsequently training the CodeT5-base model. In Table 2, we show the performance of RetroNLU compared to the original CodeT5-base. However, the experimental results indicate that the Retro method does not yield a performance boost.


# Novelty and Strengthen of Adacom (all reviewers)
###  Adapting a new model given a test sample
The current model is trained in a generalized manner to achieve overall performance on the test dataset. However, for specific individual samples, a more tailored model might be better suited to accurately predict the outcome. For large language models like GPT-4, using well-crafted prompts could serve as an effective means of instructing the model. On the other hand, for smaller models, Adacom could provide a simple approach to instruct the model for specific individual samples.

###  Use influence functions to remove noise
To train the model for a specific test sample, it is necessary to identify an appropriate small training dataset. One direct approach is to locate "similar" samples within the training data, such as using cosine similarity with the [CLS] embedding of the encoder. Nevertheless, our experimentation has shown that this method tends to introduce a significant amount of noise particularly when the most "similar" samples are still not similar.

Motivated Inference function [8], we conducted an experiment where we selectively removed certain samples from the training dataset and observed the impact on predictions. Our findings indicate that in the context of code data, a single test code may heavily rely on several training codes. Removing these specific training codes can significantly disrupt the prediction accuracy for certain test code.

The original inference function requires test sample labels and is time-consuming to calculate influence. As a result, we have developed a simplified yet highly effective variant of the inference function to assess each training sample. Consequently, we eliminate the retrieved samples with low influence scores. If no training samples remain, retraining is not performed.

###  Difference from the SOTA
The SOTA transformer-based models predominantly emphasize strategies such as augmenting the training dataset[2], introducing supplementary information with prompts[6][7], or devising alternative pre-training methods[1][3]. We hold a commendable regard for these endeavours.

However, it is equally valuable to contemplate how we can maximize the utilization of the original training dataset and the initial fine-tuned model to enhance overall model performance. Adacom offers a straightforward yet highly effective approach to enhance model performance, demonstrating the potential to surpass even larger models in terms of performance.

Note that all the related new code and results have been published on our anonymous website.
[1] CodeT5: Identifier-aware Unified Pre-trained Encoder-Decoder Models for Code Understanding and Generation
[2] CodeT5+: Open Code Large Language Models for Code Understanding and Generation
[3] CodeBERT: A Pre-Trained Model for Programming and Natural Languages
[4] Unified Pre-training for Program Understanding and Generation
[5] CodeGen: An Open Large Language Model for Code with Multi-Turn Program Synthesis
[6] RetroNLU: Retrieval Augmented Task-Oriented Semantic Parsing
[7] Retrieval Augmented Code Generation and Summarization
[8] Understanding Black-box Predictions via Influence Functions

---

> ### Author Response · Authors · 2023-08-20
>
> To all the reviewers/Area Chair/Senior Area Chair:
>
> During the rebuttal process, our discussion gravitates towards whether it is worthy to re-adapting a deep learning model on-the-fly for generating comments with higher quality. Thus, we summarise the cost and the benefit of our approach AdaCom as follows:
>
>
> - Cost: it may take extra a few seconds for the model to output comments. Note that, AdaCom incurs no extra runtime overhead if it detects that there is no need to do runtime re-adaption.
>
>
> - Benefit: comment with higher quality, especially in the out-of-distribution data (Table 2, 3, and 4 have showed this)
>
>
> We design this approach out of our industrial collaboration with XXX company (a large global company, we hide the name for anonymity). To further present the design in a more intuitive way, we show how AdaCom can work in VSCode IDE on a desktop with AMD Ryzen 7 CPU and RTX 4070 GPU as the following anonymous YouTube video.
> In the video, we show the performance of generating comment with base model (code t5, 220M) and AdaCom:
>
>
> https://www.youtube.com/watch?v=o-KpWsVkjD4
>
>
> We can see that:
>
>
> (1) *Asynchronized Design*:
>
>
> The user experience has almost no difference when waiting for over 1s (base model) or 3s (AdaCom).
>
>
> Even for the base model, it still takes some time to generate comments. Thus, we allow the user to do some other tasks while the model is generating comments. The user will be aware once the comment-generating process is done.
>
>
> (2) *Better Performance*
>
>
> With the on-the-fly adaption, the generated comment is of much higher quality.
>
>
> Note that, this work targets a cost-effective solution which exploits the best of normal models, allowing programmers with personal computers to achieve better performance.
>
>
> Last, we sincerely thank the PC, Area Chair, and Senior Area Chair for their great efforts and appreciation of our work. The anonymous reviews are helpful and valuable. We will provide a revision regarding all the comments with your valuable insight.

---

### Decision · Program_Chairs · 2023-09-21

**Decision:**

Accept (poster)

**Comment:**

The paper introduces AdaCom, a method for adapting smaller language models to improve code-comment generation, supported by metrics and experiments across multiple datasets.

The strengths are mostly aligned around the strong experimental results, systematic presentation, and the paper's potential contribution to adapting smaller models for unseen projects.

The weaknesses center on a lack of clarity in the paper (to4c, EdLg). Specific concerns about missing hyperparameters and metrics (EdLg).  Missing comparisons (to4c and EdLg). Ugvn and 4N9q focus on the issue of performance gains diminishing as model size increases, with Ugvn adding concerns about computational costs.

After considering the rebuttal and discussions, it seems like the major concerns are resolved.